# Investigating wave solutions and impact of nonlinearity: Comprehensive study of the KP-BBM model with bifurcation analysis

S. M. Rayhanul Islam[1]*, Kamruzzaman Khan[1,2]

1 Department of Mathematics, Pabna University of Science and Technology, Pabna, Bangladesh, 2 School of Science and Technology, University of New England, Armidale, NSW, Australia

* rayhanulmath@yahoo.com, rayhanul_math@pust.ac.bd

**Data Availability Statement:** The authors confirm that the data supporting the findings of this study are available within the article.

## Abstract

In this paper, we investigate the (2+1)-dimensional Kadomtsev-Petviashvili-Benjamin-Bona Mahony equation using two effective methods: the unified scheme and the advanced auxiliary equation scheme, aiming to derive precise wave solutions. These solutions are expressed as combinations of trigonometric, rational, hyperbolic, and exponential functions. Visual representations, including three-dimensional (3D) and two-dimensional (2D) combined charts, are provided for some of these solutions. The influence of the nonlinear parameter $p$ on the wave type is thoroughly examined through diverse figures, illustrating the profound impact of nonlinearity. Additionally, we briefly investigate the Hamiltonian function and the stability of the model using a planar dynamical system approach. This involves examining trajectories, isoclines, and nullclines to illustrate stable solution paths for the wave variables. Numerical results demonstrate that these methods are reliable, straightforward, and potent tools for analyzing various nonlinear evolution equations found in physics, applied mathematics, and engineering.

## 1. Introduction

In this study, we set the context by offering an overview of previous research relevant to the model under examination. By identifying gaps in the existing literature, delineating our primary objectives, and outlining the structure of this manuscript, we aim to provide a comprehensive understanding within this section.

### 1.1. Background and mathematical model

In the realm of nonlinear wave phenomena, the NLEEs find application across diverse scientific and engineering domains. These equations are presence in nonlinear wave phenomena that enable the analysis of complex occurrences in various fields. As a result, these equations have become the main tool for modelling a large number of physical phenomena in various disciplines and are frequently employed as models for a deeper understanding of complex physical phenomena. Given this, the search for exact or soliton solutions becomes crucial as it

**Funding:** The author(s) received no specific funding for this work.

**Competing interests:** Authors have no competing interest.

**Abbreviations:** NLEEs, Nonlinear evolution equations; PDE, Partial differential equation; KP-BBM, Kadomtsev-Petviashvili-Benjamin-Bona Mahony equation; AAE, Advanced auxiliary equation; GERF, Generalized exponential rational function; KP, Kadomtsev-Petviashvili; BBM, Benjamin-Bona-Mahony; KdV, KortewegDe Vries.

enables a deeper comprehension of nonlinear wave phenomena. Extracting solutions from NLEEsallows researchers to unravel the intricate dynamics and physical properties inherent in these phenomena. For this, investigators have succeeded in finding solutions for the NLEEs using a variety of analytical and numerical methods. Among many techniques, there are some efficient and powerful schemes which are the improved modified extended tanh-function [1], the new extended generalized Kudryashov [2], the unified [3], the enhanced Kudryashovs [4], the linear superposition principle and weight algorithm [5], the AAE [6], the modified extended auxiliary equation mapping [7], the Sardar sub-equation [8], the GERF and modified auxiliary equation [9], the improve $F$-expansion [10], the Hirota's bilinear [11], the modification of the simplest equation [12], the Darboux transformation [13] and numerous other approaches. Among the many techniques, our stated techniques are effective and powerful to obtain exact/soliton solutions from the NLEEs.

In our investigation, we delve into the (2+1)-dimensional KP-BBM equation, a mathematical construct that amalgamates features of the KP and BBM equations. Specifically, the KP equation is a PDE that characterizes the evolution of two-dimensional, weakly nonlinear, and weakly dispersive water waves. Originating in 1970 by BB Kadomtsev and VI Petviashvili [14], the KP equation naturally extends the KdV equation. On the other hand, the BBM equation, proposed in 1972 by JL Bona, TJB Benjamin, and JJ Mahony [15], is another nonlinear PDE modelling long waves in dispersive media, also stemming from the KdV equation. This manuscript explores the (2+1)-dimensional KP-BBM equation, a synthesis of the KP and BBM equations, with researchers continuously investigating its applications and properties within the realm of nonlinear wave phenomena across diverse scientific and engineering disciplines. The KP-BBM model [16–27] is given as

$$u_{xt} + u_{xx} + p(u^2)_{xx} + qu_{xxxt} + ru_{yy} = 0. \tag{1.1}$$

In Eq (1.1), the term $u_{xt}$ describes how the wave amplitude changes with both space $x$ and time $t$, the term $u_{xx}$ describes the curvature or spatial variations in the wave in the $x$-direction, the term $p(u^2)_{xx}$ captures the second spatial derivatives and taking into account the nonlinearity, the term $qu_{xxxt}$ describes the dispersive effect in the $x$-direction and the term $ru_{yy}$ captures the spatial variations in the $y$-direction. Overall, the terms in the equation account for changes in amplitude, spatial variations, nonlinear interactions and dispersive effects in both $x$ and $y$-directions. The coefficients $p,q$ and $r$ influence the behaviours of Eq (1.1). The wave phenomena of the model efforts bear substantial significance for the realms of fluid dynamics and shallow water waves, especially in understanding wave patterns along coastal regions and harbours. The equation serves as a mathematical instrument for describing precise wave dynamics in fluids, thereby enhancing our comprehension of nonlinear phenomena within these systems.

## 1.2. Literature review

To our knowledge, a lot of investigators have studied the KP-BBM model and explored soliton solutions through various types of techniques. In more detail, Wazwaz used the sine–cosine method, the tanh method and the extended tanh method for finding solitary wave solutions of the Eq (1.1) in Ref. [16, 17]. Abdou [18] obtained exact periodic wave solutions of Eq (1.1) by using the extended mapping method. Yu and Ma [19] have inspected the explicit solutions of Eq (1.1) through the exp-function method. Song et al. [20] have inspected the soliton solutions of the KP-BBM model with power law nonlinearity and analyzed the stability of the model. Alam and Akbar [21] have investigated the exact travelling wave solutions of Eq (1.1) by using the new approach of generalized $(G'/G)$-expansion method. Yel et al. [22] have constructed the

dark-bright soliton of the Eq (1.1) through the sine-Gordon expansion method. Manafian et al. [23] have investigated the periodic wave solutions of Eq (1.1) by using the Hirota bilinear operator method and also discuss the modulation instability of the attained solutions. Kumar et al. [24] have constructed the abundant exact solutions of Eq (1.1) by using two powerful techniques via the Lie symmetry and the GERF methods. Mia et al. [25] have inspected the novel exact travelling waves solutions of Eq (1.1) through the $(G'/G'+G+A)$-expansion technique. Tariq and Seadawy [26] have inspected the analytical soliton solutions by the auxiliary equation method. Lu et al. [27] have obtained lump solutions of Eq (1.1) through the Hirota bilinear form and also investigated interactions between lump-kink solutions and lump-soliton solutions. Given their diverse applications, these wave solutions continue to pique the interest of researchers and find utility in various fields, such as understanding wave patterns in coastal regions, harbours and others.

### 1.3. Research gap

A review of previous works on the KP-BBM model indicates that the unified (refer to S1 File) and AAE (refer to S2 File) techniques have not been utilized by other researchers. Additionally, soliton solutions have not been derived through these approaches, and the impact of the parameters has not been discussed in the existing literature. Furthermore, none of these previous authors analyzed trajectories, isoclines, and nullclines and demonstrated paths to stable solutions for the wave variable $\xi$, which none of the previous authors discussed. This observation highlights a void in the current research literature, a gap that our study seeks to fill.

### 1.4. Aim and objectives

The aims and objectives of this study are as follows: Firstly, we will discuss the stability analysis of the solutions from the stated model using the planar dynamical theory and it involves trajectories, isoclines, and nullclines to illustrate stable solution paths for the wave variable and to pinpoint the associated Hamiltonian functions. Next, we will apply the unified [3, 28] and AAE [6, 29] techniques to the stated model to explore soliton solutions from it, and also examine the influence of parameters. Additionally, we will also clarify the characteristics of the soliton pulse, offering both graphical and physical explanations within the context of the integral KP-BBM model.

### 1.5. Structure of the study

The rest of this paper is designed as follows: we have done the mathematical analysis in section 2 including applying the unified and AAE methods to the KP-BBM model and compared between our solutions and Wazwaz [16] solutions in the same section. The graphical and physical interpretation of some solutions of the KP-BBM model and the implications of parameters are also discussed in section 3. The stability analysis of the model is presented in section 4. Finally, we offered a comprehensive conclusion to summarize our findings in section 5.

## 2. Mathematical analysis

In this section, we will apply the unified [3, 28] and AAE [6, 29] schemes to the KP-BBM model for exploring the wave solutions. For this, the wave transformation is

$$u(x, t) = \varphi(\xi) \text{ and } \xi = \lambda x + \mu y - \sigma t, \tag{2.1}$$

In Eq (2.1), the coefficients $\lambda$ and $\mu$ represent the width of the soliton in $x$ and $y$-directions and $\sigma$ is the speed of the soliton. To transform the Eq (1.1) using the Eq (2.1), yields

$$q\sigma\lambda^3\varphi^{(4)} + (\sigma\lambda - r\mu^2 - \lambda^2)\varphi'' - 2p\lambda^2(\varphi\varphi'' + \varphi'^2) = 0. \qquad (2.2)$$

Integrating twice time in Eq (2.2) and integrating constant is zero, we have

$$q\sigma\lambda^3\varphi'' + (\sigma\lambda - r\mu^2 - \lambda^2)\varphi - p\lambda^2\varphi^2 = 0, \qquad (2.3)$$

Applying the homogeneous balanced principal rule in Eq (2.3) yields $N = 2$.

## 2.1. Unified scheme for KP-BBM model

After getting a balanced value, the general solution of the unified scheme [3, 28] takes the form

$$\varphi(\xi) = A_0 + A_1 w(\xi) + A_2 w^2(\xi) + B_1 w^{-1}(\xi) + B_2 w^{-2}(\xi), \qquad (2.4)$$

where $A_0, A_1, A_2, B_1$ and $B_2$ are constants and to be evaluated latter, and $w = w(\xi)$ satisfies the first order nonlinear Riccati differential equation as $w'(\xi) = w^2(\xi) + k$. Inserting Eq (2.4) into Eq (2.3) and collecting all terms of $w(\xi)$ together, equating each coefficient to zero yields a set of algebraic equations. Using Maple computation software, to solve the algebraic equations, the following solution sets are obtained:

Case 1: $\sigma = \frac{r\mu^2 + \lambda^2}{\lambda(4k\lambda^2 q + 1)}, A_0 = \frac{2qk(r\mu^2 + \lambda^2)}{(4k\lambda^2 q + 1)p}, A_1 = 0, A_2 = 0, B_1 = 0, B_2 = \frac{6q(r\mu^2 + \lambda^2)k^2}{(4k\lambda^2 q + 1)p},$

Case 2: $\sigma = -\frac{r\mu^2 + \lambda^2}{\lambda(4k\lambda^2 q - 1)}, A_0 = -\frac{6qk(r\mu^2 + \lambda^2)}{(4k\lambda^2 q - 1)p}, A_1 = 0, A_2 = 0, B_1 = 0, B_2 = -\frac{6q(r\mu^2 + \lambda^2)k^2}{(4k\lambda^2 q - 1)p},$

Case 3: $\sigma = \frac{r\mu^2 + \lambda^2}{\lambda(4k\lambda^2 q + 1)}, A_0 = \frac{2q(r\mu^2 + \lambda^2)}{(4k\lambda^2 q + 1)p}, A_1 = 0, A_2 = \frac{6q(r\mu^2 + \lambda^2)}{(4k\lambda^2 q + 1)p}, B_1 = 0, B_2 = 0,$

Case 4: $\sigma = -\frac{r\mu^2 + \lambda^2}{\lambda(4k\lambda^2 q - 1)}, A_0 = -\frac{6qk(r\mu^2 + \lambda^2)}{(4k\lambda^2 q - 1)p}, A_1 = 0, A_2 = -\frac{6q(r\mu^2 + \lambda^2)}{(4k\lambda^2 q - 1)p}, B_1 = 0, B_2 = 0,$

Inserting the above values in Eq (2.4) along with Eq (2.3), we can attain the following families of the solutions as the KP-BBM model.

**Family one:**

If $k < 0$, we obtain

$$u_1(\xi) = \frac{2qk(r\mu^2 + \lambda^2)}{(4k\lambda^2 q + 1)p} + \frac{6q(r\mu^2 + \lambda^2)k^2}{(4k\lambda^2 q + 1)p} \times \frac{(\kappa \sinh(2\sqrt{-k}(\xi + \mathcal{H})) + \ell)^2}{(\sqrt{-(\kappa^2 + \ell^2)k} - \kappa\sqrt{-k}\cosh(2\sqrt{-k}(\xi + \mathcal{H})))^2},$$

$$u_2(\xi) = \frac{2qk(r\mu^2 + \lambda^2)}{(4k\lambda^2 q + 1)p} + \frac{6q(r\mu^2 + \lambda^2)k^2}{(4k\lambda^2 q + 1)p} \times \frac{(\kappa \sinh(2\sqrt{-k}(\xi + \mathcal{H})) + \ell)^2}{(-\sqrt{-(\kappa^2 + \ell^2)k} - \kappa\sqrt{-k}\cosh(2\sqrt{-k}(\xi + \mathcal{H})))^2},$$

$$u_3(\xi) = \frac{2qk(r\mu^2 + \lambda^2)}{(4k\lambda^2 q + 1)p} + \frac{6q(r\mu^2 + \lambda^2)k^2}{(4k\lambda^2 q + 1)p} \times \frac{1}{\left(\sqrt{-k} - \frac{2\kappa\sqrt{-k}}{\kappa + \cosh(2\sqrt{-k}(\xi + \mathcal{H})) - \sinh(2\sqrt{-k}(\xi + \mathcal{H}))}\right)^2},$$

$$u_4(\xi) = \frac{2qk(r\mu^2 + \lambda^2)}{(4k\lambda^2 q + 1)p} + \frac{6q(r\mu^2 + \lambda^2)k^2}{(4k\lambda^2 q + 1)p} \times \frac{1}{\left(-\sqrt{-k} + \frac{2\kappa\sqrt{-k}}{\kappa + \cosh(2\sqrt{-k}(\xi + \mathcal{H})) + \sinh(2\sqrt{-k}(\xi + \mathcal{H}))}\right)^2},$$

if $k > 0$, we obtain

$$u_5(\xi) = \frac{2qk(r\mu^2 + \lambda^2)}{(4k\lambda^2 q + 1)p} + \frac{6q(r\mu^2 + \lambda^2)k^2}{(4k\lambda^2 q + 1)p} \times \frac{(\kappa \sin(2\sqrt{k}(\xi + \mathcal{H})) + \ell)^2}{(\sqrt{(\kappa^2 - \ell^2)k} - \kappa\sqrt{k}\cos(2\sqrt{k}(\xi + \mathcal{H})))^2},$$

$$u_6(\xi) = \frac{2qk(r\mu^2 + \lambda^2)}{(4k\lambda^2 q + 1)p} + \frac{6q(r\mu^2 + \lambda^2)k^2}{(4k\lambda^2 q + 1)p} \times \frac{(\kappa \sin(2\sqrt{k}(\xi + \mathcal{H})) + \ell)^2}{(-\sqrt{(\kappa^2 - \ell^2)k} - \kappa\sqrt{k}\cos(2\sqrt{k}(\xi + \mathcal{H})))^2},$$

$$u_7(\xi) = \frac{2qk(r\mu^2 + \lambda^2)}{(4k\lambda^2 q + 1)p} + \frac{6q(r\mu^2 + \lambda^2)k^2}{(4k\lambda^2 q + 1)p} \times \frac{1}{\left(I\sqrt{k} - \frac{2I\kappa\sqrt{k}}{\kappa + \cos(2\sqrt{k}(\xi + \mathcal{H})) - I\sin(2\sqrt{k}(\xi + \mathcal{H}))}\right)^2},$$

$$u_8(\xi) = \frac{2qk(r\mu^2 + \lambda^2)}{(4k\lambda^2 q + 1)p} + \frac{6q(r\mu^2 + \lambda^2)k^2}{(4k\lambda^2 q + 1)p} \times \frac{1}{\left(-I\sqrt{k} + \frac{2I\kappa\sqrt{k}}{\kappa + \cos(2\sqrt{k}(\xi + \mathcal{H})) + \sin(2\sqrt{k}(\xi + \mathcal{H}))}\right)^2},$$

Where $\xi = \lambda x + \mu y - \sigma t$ and $\sigma = \frac{r\mu^2 + \lambda^2}{\lambda(4k\lambda^2 q + 1)}$. All the above solutions will exist if the condition $p, q, \lambda \neq 0$ must hold.

**Family two:**

If $k < 0$, we obtain

$$u_9(\xi) = -\frac{6qk(r\mu^2 + \lambda^2)}{(4k\lambda^2 q - 1)p} - \frac{6q(r\mu^2 + \lambda^2)k^2}{(4k\lambda^2 q - 1)p} \times \frac{(\kappa \sinh(2\sqrt{-k}(\xi + \mathcal{H})) + \ell)^2}{(\sqrt{-(\kappa^2 + \ell^2)k} - \kappa\sqrt{-k}\cosh(2\sqrt{-k}(\xi + \mathcal{H})))^2},$$

$$u_{10}(\xi) = -\frac{6qk(r\mu^2 + \lambda^2)}{(4k\lambda^2 q - 1)p} - \frac{6q(r\mu^2 + \lambda^2)k^2}{(4k\lambda^2 q - 1)p} \times \frac{(\kappa \sinh(2\sqrt{-k}(\xi + \mathcal{H})) + \ell)^2}{(-\sqrt{-(\kappa^2 + \ell^2)k} - \kappa\sqrt{-k}\cosh(2\sqrt{-k}(\xi + \mathcal{H})))^2},$$

$$u_{11}(\xi) = -\frac{6qk(r\mu^2 + \lambda^2)}{(4k\lambda^2 q - 1)p} - \frac{6q(r\mu^2 + \lambda^2)k^2}{(4k\lambda^2 q - 1)p} \times \frac{1}{\left(\sqrt{-k} - \frac{2\kappa\sqrt{-k}}{\kappa + \cosh(2\sqrt{-k}(\xi + \mathcal{H})) - \sinh(2\sqrt{-k}(\xi + \mathcal{H}))}\right)^2},$$

$$u_{12}(\xi) = -\frac{6qk(r\mu^2 + \lambda^2)}{(4k\lambda^2 q - 1)p} - \frac{6q(r\mu^2 + \lambda^2)k^2}{(4k\lambda^2 q - 1)p} \times \frac{1}{\left(-\sqrt{-k} + \frac{2\kappa\sqrt{-k}}{\kappa + \cosh(2\sqrt{-k}(\xi + \mathcal{H})) + \sinh(2\sqrt{-k}(\xi + \mathcal{H}))}\right)^2},$$

if $k > 0$, we obtain

$$u_{13}(\xi) = -\frac{6qk(r\mu^2 + \lambda^2)}{(4k\lambda^2 q - 1)p} - \frac{6q(r\mu^2 + \lambda^2)k^2}{(4k\lambda^2 q - 1)p} \times \frac{(\kappa \sin(2\sqrt{k}(\xi + \mathcal{H})) + \ell)^2}{(\sqrt{(\kappa^2 - \ell^2)k} - \kappa\sqrt{k}\cos(2\sqrt{k}(\xi + \mathcal{H})))^2},$$

$$u_{14}(\xi) = -\frac{6qk(r\mu^2 + \lambda^2)}{(4k\lambda^2 q - 1)p} - \frac{6q(r\mu^2 + \lambda^2)k^2}{(4k\lambda^2 q - 1)p} \times \frac{(\kappa \sin(2\sqrt{k}(\xi + \mathcal{H})) + \ell)^2}{(-\sqrt{(\kappa^2 - \ell^2)k} - \kappa\sqrt{k}\cos(2\sqrt{k}(\xi + \mathcal{H})))^2},$$

$$u_{15}(\xi) = -\frac{6qk(r\mu^2 + \lambda^2)}{(4k\lambda^2 q - 1)p} - \frac{6q(r\mu^2 + \lambda^2)k^2}{(4k\lambda^2 q - 1)p} \times \frac{1}{\left(I\sqrt{k} - \frac{2I\kappa\sqrt{k}}{\kappa + \cos(2\sqrt{k}(\xi + \mathcal{H})) - I\sin(2\sqrt{k}(\xi + \mathcal{H}))}\right)^2},$$

$$u_{16}(\xi) = -\frac{6qk(r\mu^2 + \lambda^2)}{(4k\lambda^2 q - 1)p} - \frac{6q(r\mu^2 + \lambda^2)k^2}{(4k\lambda^2 q - 1)p} \times \frac{1}{\left(-I\sqrt{k} + \frac{2I\kappa\sqrt{k}}{\kappa + \cos(2\sqrt{k}(\xi + \mathcal{H})) + I\sin(2\sqrt{k}(\xi + \mathcal{H}))}\right)^2},$$

Where $\xi = \lambda x + \mu y - \sigma t$ and $\sigma = -\frac{r\mu^2 + \lambda^2}{\lambda(4k\lambda^2 q - 1)}$. All the above solutions will exist if the condition $p, q, \lambda \neq 0$ must hold.

**Family three:**

If $k<0$, we obtain

$$u_{17}(\xi) = \frac{2qk(r\mu^2 + \lambda^2)}{(4k\lambda^2 q + 1)p} + \frac{6q(r\mu^2 + \lambda^2)}{(4k\lambda^2 q + 1)p} \times \frac{\left(\sqrt{-(\kappa^2 + \ell^2)k} - \kappa\sqrt{-k}cosh(2\sqrt{-k}(\xi + \mathcal{H}))\right)^2}{\left(\kappa sinh(2\sqrt{-k}(\xi + \mathcal{H})) + \ell\right)^2},$$

$$u_{18}(\xi) = \frac{2qk(r\mu^2 + \lambda^2)}{(4k\lambda^2 q + 1)p} + \frac{6q(r\mu^2 + \lambda^2)}{(4k\lambda^2 q + 1)p} \times \frac{\left(-\sqrt{-(\kappa^2 + \ell^2)k} - \kappa\sqrt{-k}cosh(2\sqrt{-k}(\xi + \mathcal{H}))\right)^2}{\left(\kappa sinh(2\sqrt{-k}(\xi + \mathcal{H})) + \ell\right)^2},$$

$$u_{19}(\xi) = \frac{2qk(r\mu^2 + \lambda^2)}{(4k\lambda^2 q + 1)p} + \frac{6q(r\mu^2 + \lambda^2)}{(4k\lambda^2 q + 1)p} \times \left(\sqrt{-k} - \frac{2\kappa\sqrt{-k}}{\kappa + cosh(2\sqrt{-k}(\xi + \mathcal{H})) - sinh(2\sqrt{-k}(\xi + \mathcal{H}))}\right)^2,$$

$$u_{20}(\xi) = \frac{2qk(r\mu^2 + \lambda^2)}{(4k\lambda^2 q + 1)p} + \frac{6q(r\mu^2 + \lambda^2)}{(4k\lambda^2 q + 1)p} \times \left(-\sqrt{-k} + \frac{2\kappa\sqrt{-k}}{\kappa + cosh(2\sqrt{-k}(\xi + \mathcal{H})) + sinh(2\sqrt{-k}(\xi + \mathcal{H}))}\right)^2,$$

if $k>0$, we obtain

$$u_{21}(\xi) = \frac{2qk(r\mu^2 + \lambda^2)}{(4k\lambda^2 q + 1)p} + \frac{6q(r\mu^2 + \lambda^2)}{(4k\lambda^2 q + 1)p} \times \frac{\left(\sqrt{(\kappa^2 - \ell^2)k} - \kappa\sqrt{k}cos(2\sqrt{k}(\xi + \mathcal{H}))\right)^2}{\left(\kappa sin(2\sqrt{k}(\xi + \mathcal{H})) + \ell\right)^2},$$

$$u_{22}(\xi) = \frac{2qk(r\mu^2 + \lambda^2)}{(4k\lambda^2 q + 1)p} + \frac{6q(r\mu^2 + \lambda^2)}{(4k\lambda^2 q + 1)p} \times \frac{\left(-\sqrt{(\kappa^2 - \ell^2)k} - \kappa\sqrt{k}cos(2\sqrt{k}(\xi + \mathcal{H}))\right)^2}{\left(\kappa sin(2\sqrt{k}(\xi + \mathcal{H})) + \ell\right)^2},$$

$$u_{23}(\xi) = \frac{2qk(r\mu^2 + \lambda^2)}{(4k\lambda^2 q + 1)p} + \frac{6q(r\mu^2 + \lambda^2)}{(4k\lambda^2 q + 1)p} \times \left(I\sqrt{k} - \frac{2I\kappa\sqrt{k}}{\kappa + cos(2\sqrt{k}(\xi + \mathcal{H})) - Isin(2\sqrt{k}(\xi + \mathcal{H}))}\right)^2,$$

$$u_{24}(\xi) = \frac{2qk(r\mu^2 + \lambda^2)}{(4k\lambda^2 q + 1)p} + \frac{6q(r\mu^2 + \lambda^2)}{(4k\lambda^2 q + 1)p} \times \left(-I\sqrt{k} + \frac{2I\kappa\sqrt{k}}{\kappa + cos(2\sqrt{k}(\xi + \mathcal{H})) + Isin(2\sqrt{k}(\xi + \mathcal{H}))}\right)^2,$$

Where $\xi = \lambda x + \mu y - \sigma t$ and $\sigma = \frac{r\mu^2 + \lambda^2}{\lambda(4k\lambda^2 q + 1)}$. All the above solutions will exist if the condition $p, q, \lambda \neq 0$ must hold.

**Family four:**

If $k<0$, we obtain

$$u_{25}(\xi) = -\frac{6qk(r\mu^2 + \lambda^2)}{(4k\lambda^2 q - 1)p} - \frac{6q(r\mu^2 + \lambda^2)}{(4k\lambda^2 q - 1)p} \times \frac{\left(\sqrt{-(\kappa^2 + \ell^2)k} - \kappa\sqrt{-k}cosh(2\sqrt{-k}(\xi + \mathcal{H}))\right)^2}{\left(\kappa sinh(2\sqrt{-k}(\xi + \mathcal{H})) + \ell\right)^2},$$

$$u_{26}(\xi) = -\frac{6qk(r\mu^2 + \lambda^2)}{(4k\lambda^2 q - 1)p} - \frac{6q(r\mu^2 + \lambda^2)}{(4k\lambda^2 q - 1)p} \times \frac{\left(-\sqrt{-(\kappa^2 + \ell^2)k} - \kappa\sqrt{-k}cosh(2\sqrt{-k}(\xi + \mathcal{H}))\right)^2}{\left(\kappa sinh(2\sqrt{-k}(\xi + \mathcal{H})) + \ell\right)^2},$$

$$u_{27}(\xi) = -\frac{6qk(r\mu^2 + \lambda^2)}{(4k\lambda^2 q - 1)p} - \frac{6q(r\mu^2 + \lambda^2)}{(4k\lambda^2 q - 1)p} \times \left(\sqrt{-k} - \frac{2\kappa\sqrt{-k}}{\kappa + cosh(2\sqrt{-k}(\xi + \mathcal{H})) - sinh(2\sqrt{-k}(\xi + \mathcal{H}))}\right)^2,$$

$$u_{28}(\xi) = -\frac{6qk(r\mu^2+\lambda^2)}{(4k\lambda^2q-1)p} - \frac{6q(r\mu^2+\lambda^2)}{(4k\lambda^2q-1)p} \times \left( -\sqrt{-k} + \frac{2\kappa\sqrt{-k}}{\kappa+cosh(2\sqrt{-k}(\xi+\mathcal{H}))+sinh(2\sqrt{-k}(\xi+\mathcal{H}))} \right)^2,$$

if $k>0$, we obtain

$$u_{29}(\xi) = -\frac{6qk(r\mu^2+\lambda^2)}{(4k\lambda^2q-1)p} - \frac{6q(r\mu^2+\lambda^2)}{(4k\lambda^2q-1)p} \times \frac{(\sqrt{(\kappa^2-\ell^2)k}-\kappa\sqrt{k}cos(2\sqrt{k}(\xi+\mathcal{H})))^2}{(\kappa sin(2\sqrt{k}(\xi+\mathcal{H}))+\ell)^2},$$

$$u_{30}(\xi) = -\frac{6qk(r\mu^2+\lambda^2)}{(4k\lambda^2q-1)p} - \frac{6q(r\mu^2+\lambda^2)}{(4k\lambda^2q-1)p} \times \frac{(-\sqrt{(\kappa^2-\ell^2)k}-\kappa\sqrt{k}cos(2\sqrt{k}(\xi+\mathcal{H})))^2}{(\kappa sin(2\sqrt{k}(\xi+\mathcal{H}))+\ell)^2},$$

$$u_{31}(\xi) = -\frac{6qk(r\mu^2+\lambda^2)}{(4k\lambda^2q-1)p} - \frac{6q(r\mu^2+\lambda^2)}{(4k\lambda^2q-1)p} \times \left( I\sqrt{k} - \frac{2I\kappa\sqrt{k}}{\kappa+cos(2\sqrt{k}(\xi+\mathcal{H}))-Isin(2\sqrt{k}(\xi+\mathcal{H}))} \right)^2,$$

$$u_{32}(\xi) = -\frac{6qk(r\mu^2+\lambda^2)}{(4k\lambda^2q-1)p} - \frac{6q(r\mu^2+\lambda^2)}{(4k\lambda^2q-1)p} \times \left( -I\sqrt{k} + \frac{2I\kappa\sqrt{k}}{\kappa+cos(2\sqrt{k}(\xi+\mathcal{H}))+Isin(2\sqrt{k}(\xi+\mathcal{H}))} \right)^2,$$

Where $\xi = \lambda x + \mu y - \sigma t$ and $\sigma = -\frac{r\mu^2+\lambda^2}{\lambda(4k\lambda^2q-1)}$. All the above solutions will exist if the condition $p, q, \lambda \neq 0$ must hold. If $k = 0$, there is no solution of the Eq (1.1).

## 2.2. AAE scheme for KP-BBM model

After getting a balanced value, the general solution of the AAE [6, 29] scheme takes the form

$$\varphi(\xi) = c_0 + c_1 a^{g(\xi)} + c_2 a^{2g(\xi)}, \tag{2.5}$$

Where $c_0, c_1$ and $c_2 (\neq 0)$ are constants and to be evaluated later, and $g = g(\xi)$ satisfies the first-order nonlinear Riccati differential equation as $g'(\xi) = \frac{1}{\ln(a)}\{\alpha a^{-g(\xi)} + \beta + \gamma a^{2g(\xi)}\}$. By substituting Eq (2.5) into Eq (2.3), we derive a set of algebraic equations, which upon solution, provide the following solution sets:

Case 1:

$\sigma = -\frac{r\mu^2+\lambda^2}{\lambda(4\alpha\gamma\lambda^2q-\beta^2\lambda^2q-1)}, c_0 = -\frac{2\alpha q(r\mu^2+\lambda^2)\gamma}{(4\alpha\gamma\lambda^2q-\beta^2\lambda^2q-1)p}, c_1 = -\frac{6\beta\gamma q(r\mu^2+\lambda^2)}{(4\alpha\gamma\lambda^2q-\beta^2\lambda^2q-1)p}, c_2 = -\frac{6\gamma^2q(r\mu^2+\lambda^2)}{(4\alpha\gamma\lambda^2q-\beta^2\lambda^2q-1)p},$

Case 2:

$\sigma = \frac{r\mu^2+\lambda^2}{\lambda(4\alpha\gamma\lambda^2q-\beta^2\lambda^2q+1)}, c_0 = \frac{q(2\alpha r\mu^2 r+\beta^2\mu^2 r+2\alpha\gamma\lambda^2+\beta^2\lambda^2)}{(4\alpha\gamma\lambda^2q-\beta^2\lambda^2q+1)p}, c_1 = \frac{6\beta\gamma q(r\mu^2+\lambda^2)}{(4\alpha\gamma\lambda^2q-\beta^2\lambda^2q+1)p}, c_2 = \frac{6\gamma^2q(r\mu^2+\lambda^2)}{(4\alpha\gamma\lambda^2q-\beta^2\lambda^2q+1)p}.$

Inserting the above values in Eq (2.5) along with Eq (2.3), we can attain the following families of the solutions as the KP-BBM model.

**Family one:**

When $\beta^2-4\alpha\gamma<0$ and $\gamma \neq 0$,

$$u_{33}(\xi) = -\frac{3q(r\mu^2+\lambda^2)(\beta^2-4\alpha\gamma)}{2p(1+q\lambda^2(\beta^2-4\alpha\gamma))} \times \left( tan^2\left( \frac{\sqrt{4\alpha\gamma-\beta^2}}{2}\xi \right) + 1 \right),$$

and

$$u_{34}(\xi) = -\frac{3q(r\mu^2 + \lambda^2)(\beta^2 - 4\alpha\gamma)}{2p(1 + q\lambda^2(\beta^2 - 4\alpha\gamma))} \times \left(cot^2\left(\frac{\sqrt{4\alpha\gamma - \beta^2}}{2}\xi\right) + 1\right).$$

When $\beta^2 - 4\alpha\gamma > 0$ and $\gamma \neq 0$,

$$u_{35}(\xi) = -\frac{3q(r\mu^2 + \lambda^2)(\beta^2 - 4\alpha\gamma)}{2p(1 + q\lambda^2(\beta^2 - 4\alpha\gamma))} \operatorname{sech}^2\left(\frac{\sqrt{\beta^2 - 4\alpha\gamma}}{2}\xi\right),$$

and

$$u_{36}(\xi) = \frac{3q(r\mu^2 + \lambda^2)(\beta^2 - 4\alpha\gamma)}{2p(1 + q\lambda^2(\beta^2 - 4\alpha\gamma))} \operatorname{csch}^2\left(\frac{\sqrt{\beta^2 - 4\alpha\gamma}}{2}\xi\right).$$

When $\beta^2 - 4\alpha^2 < 0, \gamma \neq 0$ and $\gamma = -\alpha$,

$$u_{37}(\xi) = -\frac{3q(r\mu^2 + \lambda^2)(\beta^2 + 4\alpha^2)}{2p(1 + q\lambda^2(\beta^2 + 4\alpha^2))} \sec^2\left(\frac{\sqrt{-4\alpha^2 - \beta^2}}{2}\xi\right),$$

and

$$u_{38}(\xi) = -\frac{3q(r\mu^2 + \lambda^2)(\beta^2 + 4\alpha^2)}{2p(1 + q\lambda^2(\beta^2 + 4\alpha^2))} \times csc^2\left(\frac{\sqrt{-4\alpha^2 - \beta^2}}{2}\xi\right).$$

When $\beta^2 - 4\alpha^2 > 0, \gamma \neq 0$ and $\gamma = -\alpha$,

$$u_{39}(\xi) = \frac{3q(r\mu^2 + \lambda^2)(\beta^2 + 4\alpha^2)}{2p(1 + q\lambda^2(\beta^2 + 4\alpha^2))} \times \left(tanh^2\left(\frac{\sqrt{4\alpha^2 + \beta^2}}{2}\xi\right) - 1\right),$$

and

$$u_{40}(\xi) = \frac{3q(r\mu^2 + \lambda^2)(\beta^2 + 4\alpha^2)}{2p(1 + q\lambda^2(\beta^2 + 4\alpha^2))} \times \left(coth^2\left(\frac{\sqrt{4\alpha^2 + \beta^2}}{2}\xi\right) - 1\right).$$

When $\beta^2 - 4\alpha^2 < 0$ and $\gamma = \alpha$,

$$u_{41}(\xi) = -\frac{3q(r\mu^2 + \lambda^2)(\beta^2 - 4\alpha^2)}{2p(1 + q\lambda^2(\beta^2 - 4\alpha^2))} \times \left(tan^2\left(\frac{\sqrt{4\alpha^2 - \beta^2}}{2}\xi\right) + 1\right),$$

and

$$u_{42}(\xi) = -\frac{3q(r\mu^2 + \lambda^2)(\beta^2 - 4\alpha^2)}{2p(1 + q\lambda^2(\beta^2 - 4\alpha^2))} \times \left(cot^2\left(\frac{\sqrt{4\alpha^2 - \beta^2}}{2}\xi\right) + 1\right).$$

When $\beta^2 - 4\alpha^2 > 0$ and $\gamma = \alpha$,

$$u_{43}(\xi) = \frac{3q(r\mu^2 + \lambda^2)(\beta^2 - 4\alpha^2)}{2p(1 + q\lambda^2(\beta^2 - 4\alpha^2))} \times \left(tanh^2\left(\frac{\sqrt{\beta^2 - 4\alpha^2}}{2}\xi\right) - 1\right),$$

and

$$u_{44}(\xi) = \frac{3q(r\mu^2 + \lambda^2)(\beta^2 - 4\alpha^2)}{2p(1 + q\lambda^2(\beta^2 - 4\alpha^2))} \times \left( coth^2\left( \frac{\sqrt{\beta^2 - 4\alpha^2}}{2} \xi \right) - 1 \right).$$

When $\beta^2 = 4\alpha\gamma$,

$$u_{45}(\xi) = \frac{6q(r\mu^2 + \lambda^2)}{p\xi^2}.$$

When $\alpha\gamma < 0, \beta = 0$ and $\gamma \neq 0$,

$$u_{46}(\xi) = -\frac{6\alpha q(r\mu^2 + \lambda^2)\gamma}{p(4\alpha\gamma\lambda^2 q - 1)} \times sech^2(\sqrt{-\alpha\gamma}\xi),$$

and

$$u_{47}(\xi) = \frac{6\alpha q(r\mu^2 + \lambda^2)\gamma}{p(4\alpha\gamma\lambda^2 q - 1)} \times cosech^2(\sqrt{-\alpha\gamma}\xi).$$

When $\beta = 0$ and $\alpha = -\gamma$,

$$u_{48}(\xi) = \frac{24q\gamma^2(r\mu^2 + \lambda^2)}{p(4\gamma^2\lambda^2 q + 1)} \times \frac{e^{-2\gamma\xi}}{(e^{-2\gamma\xi} - 1)^2}.$$

When $\beta = \gamma = K$ and $\alpha = 0$,

$$u_{49}(\xi) = \frac{6qK^2(r\mu^2 + \lambda^2)}{p(K^2\lambda^2 q + 1)} \times \frac{e^{K\xi}}{(e^{K\xi} - 1)^2}.$$

When $\beta = (\alpha + \gamma)$,

$$u_{50}(\xi) = \frac{6q\gamma(r\mu^2 + \lambda^2)}{p + pq\lambda^2(\alpha - \gamma)^2} \times \frac{(\alpha - \gamma)^2 e^{(\alpha - \gamma)\xi}}{(\gamma e^{(\alpha - \gamma)\xi} - 1)^2}.$$

When $\beta = -(\alpha + \gamma)$,

$$u_{51}(\xi) = \frac{6q\gamma(r\mu^2 + \lambda^2)}{p + pq\lambda^2(\alpha - \gamma)^2} \times \frac{(\alpha - \gamma)^2 e^{(\alpha - \gamma)\xi}}{(\gamma - e^{(\alpha - \gamma)\xi})^2}.$$

When $\alpha = 0$,

$$u_{52}(\xi) = \frac{6q\gamma(r\mu^2 + \lambda^2)}{p + pq\lambda^2\beta^2} \times \frac{\beta^2 e^{\beta\xi}}{(\gamma e^{\beta\xi} - 1)^2}.$$

When $\gamma = \beta = \alpha \neq 0$,

$$u_{53}(\xi) = -\frac{3q\alpha^2(r\mu^2 + \lambda^2)}{2p(\alpha^2\lambda^2 q - \frac{1}{3})} \times \left( tan^2\left( \frac{\sqrt{3}\alpha}{2} \xi \right) + 1 \right).$$

When $\alpha = \beta = 0$,

$$u_{54}(\xi) = \frac{6q(r\mu^2 + \lambda^2)}{p\xi^2}.$$

When $\gamma = \alpha$ and $\beta = 0$,

$$u_{55}(\xi) = -\frac{6q\alpha^2(r\mu^2 + \lambda^2)}{p(4\alpha^2\lambda^2 q - 1)} \times sec^2(\alpha\xi).$$

Where $\xi = \lambda x + \mu y - \sigma t$ and $\sigma = -\frac{r\mu^2 + \lambda^2}{\lambda(4\alpha\gamma\lambda^2 q - \beta^2\lambda^2 q - 1)}$. Under specific conditions, such as $\gamma = \alpha = 0$, $\alpha = \beta = K$ and $\gamma = 0$, the solution of the KP-BBM model does not exist. All the above solutions will exist if the condition $p,q,\lambda \neq 0$ must hold.

## Family two

When $\beta^2 - 4\alpha\gamma < 0$ and $\gamma \neq 0$,

$$u_{56}(\xi) = \frac{3q(r\mu^2 + \lambda^2)(\beta^2 - 4\alpha\gamma)}{2p(-1 + q\lambda^2(\beta^2 - 4\alpha\gamma))} \times \left( tan^2\left(\frac{\sqrt{4\alpha\gamma - \beta^2}}{2}\xi\right) + \frac{1}{3} \right),$$

and

$$u_{57}(\xi) = \frac{3q(r\mu^2 + \lambda^2)(\beta^2 - 4\alpha\gamma)}{2p(-1 + q\lambda^2(\beta^2 - 4\alpha\gamma))} \times \left( cot^2\left(\frac{\sqrt{4\alpha\gamma - \beta^2}}{2}\xi\right) + \frac{1}{3} \right).$$

When $\beta^2 - 4\alpha\gamma > 0$ and $\gamma \neq 0$,

$$u_{58}(\xi) = -\frac{3q(r\mu^2 + \lambda^2)(\beta^2 - 4\alpha\gamma)}{2p(-1 + q\lambda^2(\beta^2 - 4\alpha\gamma))} \times \left( tanh^2\left(\frac{\sqrt{\beta^2 - 4\alpha\gamma}}{2}\xi\right) - \frac{1}{3} \right),$$

and

$$u_{59}(\xi) = -\frac{3q(r\mu^2 + \lambda^2)(\beta^2 - 4\alpha\gamma)}{2p(-1 + q\lambda^2(\beta^2 - 4\alpha\gamma))} \times \left( coth^2\left(\frac{\sqrt{\beta^2 - 4\alpha\gamma}}{2}\xi\right) - \frac{1}{3} \right).$$

When $\beta^2 + 4\alpha^2 < 0, \gamma \neq 0$ and $\gamma = -\alpha$,

$$u_{60}(\xi) = \frac{3q(r\mu^2 + \lambda^2)(\beta^2 + 4\alpha^2)}{2p(-1 + q\lambda^2(\beta^2 + 4\alpha^2))} \times \left( tan^2\left(\frac{\sqrt{-4\alpha^2 - \beta^2}}{2}\xi\right) + \frac{1}{3} \right),$$

and

$$u_{61}(\xi) = \frac{3q(r\mu^2 + \lambda^2)(\beta^2 + 4\alpha^2)}{2p(1 + q\lambda^2(\beta^2 + 4\alpha^2))} \times \left( cot^2\left(\frac{\sqrt{-4\alpha^2 - \beta^2}}{2}\xi\right) + \frac{1}{3} \right).$$

When $\beta^2 + 4\alpha^2 > 0, \gamma \neq 0$ and $\gamma = -\alpha$,

$$u_{62}(\xi) = -\frac{3q(r\mu^2 + \lambda^2)(\beta^2 + 4\alpha^2)}{2p(-1 + q\lambda^2(\beta^2 + 4\alpha^2))} \times \left( tanh^2\left(\frac{\sqrt{4\alpha^2 + \beta^2}}{2}\xi\right) - \frac{1}{3} \right),$$

and

$$u_{63}(\xi) = -\frac{3q(r\mu^2 + \lambda^2)(\beta^2 + 4\alpha^2)}{2p(-1 + q\lambda^2(\beta^2 + 4\alpha^2))} \times \left( coth^2\left(\frac{\sqrt{4\alpha^2 + \beta^2}}{2}\xi\right) - \frac{1}{3}\right).$$

When $\beta^2 - 4\alpha^2 < 0$ and $\gamma = \alpha$,

$$u_{64}(\xi) = \frac{3q(r\mu^2 + \lambda^2)(\beta^2 - 4\alpha^2)}{2p(-1 + q\lambda^2(\beta^2 - 4\alpha^2))} \times \left( tan^2\left(\frac{\sqrt{4\alpha^2 - \beta^2}}{2}\xi\right) + \frac{1}{3}\right),$$

and

$$u_{65}(\xi) = \frac{3q(r\mu^2 + \lambda^2)(\beta^2 - 4\alpha^2)}{2p(-1 + q\lambda^2(\beta^2 - 4\alpha^2))} \times \left( cot^2\left(\frac{\sqrt{4\alpha^2 - \beta^2}}{2}\xi\right) + \frac{1}{3}\right).$$

When $\beta^2 - 4\alpha^2 > 0$ and $\gamma = \alpha$,

$$u_{66}(\xi) = -\frac{3q(r\mu^2 + \lambda^2)(\beta^2 - 4\alpha^2)}{2p(-1 + q\lambda^2(\beta^2 - 4\alpha^2))} \times \left( tanh^2\left(\frac{\sqrt{\beta^2 - 4\alpha^2}}{2}\xi\right) - \frac{1}{3}\right),$$

and

$$u_{67}(\xi) = -\frac{3q(r\mu^2 + \lambda^2)(\beta^2 - 4\alpha^2)}{2p(-1 + q\lambda^2(\beta^2 - 4\alpha^2))} \times \left( coth^2\left(\frac{\sqrt{\beta^2 - 4\alpha^2}}{2}\xi\right) - \frac{1}{3}\right).$$

When $\beta^2 = 4\alpha\gamma$,

$$u_{68}(\xi) = \frac{6q(r\mu^2 + \lambda^2)}{p\xi^2}.$$

When $\alpha\gamma < 0, \beta = 0$ and $\gamma \neq 0$,

$$u_{69}(\xi) = -\frac{4\alpha\gamma q(r\mu^2 + \lambda^2)}{p(1 + 4\alpha\gamma q\lambda^2)} \times \frac{\left(cosh^2\left(\sqrt{-\alpha\gamma}\xi\right) - \frac{3}{2}\right)}{cosh^2(\sqrt{-\alpha\gamma}\xi)},$$

and

$$u_{70}(\xi) = -\frac{4\alpha\gamma q(r\mu^2 + \lambda^2)}{p(1 + 4\alpha\gamma q\lambda^2)} \times \frac{\left(cosh^2\left(\sqrt{-\alpha\gamma}\xi\right) + \frac{1}{2}\right)}{sinh^2(\sqrt{-\alpha\gamma}\xi)}.$$

When $\beta = 0$ and $\alpha = -\gamma$,

$$u_{71}(\xi) = -\frac{4q\gamma^2(r\mu^2 + \lambda^2)}{p(4\gamma^2\lambda^2 q - 1)} \times \frac{e^{-4\gamma\xi} + 4e^{-2\gamma\xi} + 1}{\left(e^{-2\gamma\xi} - 1\right)^2}.$$

When $\beta = \gamma = K$ and $\alpha = 0$,

$$u_{72}(\xi) = -\frac{qK^2(r\mu^2 + \lambda^2)}{p(K^2\lambda^2 q - 1)} \times \frac{e^{2K\xi} + 4e^{K\xi} + 1}{\left(e^{K\xi} - 1\right)^2}.$$

When $\beta = (\alpha+\gamma)$,

$$u_{73}(\xi) = -\frac{q(\alpha-\gamma)^2(r\mu^2+\lambda^2)}{pq\lambda^2(\alpha-\gamma)^2-p} \times \frac{\gamma^2 e^{2(\alpha-\gamma)\xi}+4\gamma e^{(\alpha-\gamma)\xi}+1}{(\gamma e^{(\alpha-\gamma)\xi}-1)^2}.$$

When $\beta = -(\alpha+\gamma)$,

$$u_{74}(\xi) == -\frac{q(\alpha-\gamma)^2(r\mu^2+\lambda^2)}{pq\lambda^2(\alpha-\gamma)^2-p} \times \frac{e^{2(\alpha-\gamma)\xi}+4\gamma e^{(\alpha-\gamma)\xi}+\gamma^2}{(\gamma-e^{(\alpha-\gamma)\xi})^2}.$$

When $\alpha = 0$,

$$u_{75}(\xi) = -\frac{q\beta^2(r\mu^2+\lambda^2)}{pq\lambda^2\beta^2-p} \times \frac{\gamma^2 e^{2\beta\xi}+4\gamma e^{\beta\xi}+1}{(\gamma e^{\beta\xi}-1)^2}.$$

When $\gamma = \beta = \alpha \neq 0$,

$$u_{76}(\xi) = \frac{3q\alpha^2(r\mu^2+\lambda^2)}{2p(\alpha^2\lambda^2 q+\frac{1}{3})} \times \left(tan^2\left(\frac{\sqrt{3}\alpha}{2}\xi\right)+\frac{1}{3}\right).$$

When $\alpha = \beta = 0$,

$$u_{77}(\xi) = \frac{6q(r\mu^2+\lambda^2)}{p\xi^2}.$$

When $\gamma = \alpha$ and $\beta = 0$,

$$u_{78}(\xi) = -\frac{4q\alpha^2(r\mu^2+\lambda^2)}{p(4\alpha^2\lambda^2 q+1)} \times \frac{(cos^2(\alpha\xi)-\frac{3}{2})}{cos^2(\alpha\xi)}.$$

Where $\xi = \lambda x+\mu y-\sigma t$ and $\sigma = \frac{r\mu^2+\lambda^2}{\lambda(4\alpha\gamma\lambda^2 q-\beta^2\lambda^2 q+1)}$. All the above solutions will exist if the condition $p,q,\lambda \neq 0$ must hold. It is mentioned that the following conditions, when $\gamma = \alpha = 0, \gamma = 0$ and $\alpha = \beta = K$, and $\gamma = 0$, we attained the constant solutions of the KP-BBM model. Therefore, the constant solutions lack any physical significance. On the other hand, the obtained solution of the KP-BBM model does not exist, when $\gamma = \beta = 0$. It is also mentioned that the solutions $u_{66}(\xi)$ and $u_{77}(\xi)$ are identical to the stated model, when the different conditions as $\beta^2 = 4\alpha\gamma$ and $\alpha = \beta = 0$.

## 2.3. Comparison

Wazwaz [16] discovered four precise solutions to the KP-BBM equation through the application of the sine-cosine method. In contrast, the AAE method yields numerous wave solutions for the established KP-BBM equation. It is noteworthy that both methods share a common solution, as illustrated in Table 1. Ultimately, it can be asserted that employing the AAE method for solving the KP-BBM equation results in a significantly greater number of wave solutions compared to the sine-cosine method employed by Wazwaz [16].

Utilizing the tanh method, Wazwaz [16] successfully derived merely four precise solutions for the KP-BBM equation, as elaborated in [16]. In contrast, employing the AAE method led to the discovery of forty-six wave solutions for the aforementioned equation. These solutions are expressed through exponential function solutions, rational function solutions, hyperbolic function solutions, and trigonometric function solutions. It is worth mentioning that this method also gives some common solutions, as illustrated in Table 2. Consequently, the AAE

**Table 1. Comparison of our solutions and Wazwaz [16] solutions by sine-cosine scheme.**

| Wazwaz [16] solutions by sine-cosine method | Our solutions by the AAE method |
|---|---|
| Taking $a = -\frac{7}{24}, b = \frac{1}{4}, r = 1, c = 4$ and $u(x,y,t) = \Phi(x,y,t)$, then the solution of Eq (27) turns to $$\Phi(x,y,t) = \frac{72}{7} sec^2\left(\frac{i(x+y-4t)}{\sqrt{2}}\right).$$ | Taking $\lambda = 1, \mu = 1, \gamma = 1, p = 1, r = -9, q = 3, \beta = 1, \alpha = \frac{1}{2}$ and $u_{37}(x,y,t) = \Phi(x,y,t)$, then the solution turns to $$\Phi(x,y,t) = \frac{72}{7} sec^2\left(\frac{i(x+y-4t)}{\sqrt{2}}\right).$$ |
| Picking $a = -\frac{7}{24}, b = \frac{1}{4}, r = 1, c = 4$ and $u(x,y,t) = \Phi(x,y,t)$, then the solution of Eq (26) turns to $$\Phi(x,y,t) = \frac{72}{7} csc^2\left(\frac{i(x+y-4t)}{\sqrt{2}}\right).$$ | Picking $\lambda = 1, \mu = 1, \gamma = 1, p = 1, r = -9, q = 3, \beta = 1, \alpha = \frac{1}{2}$ and $u_{38}(x,y,t) = \Phi(x,y,t)$, then the solution turns to $$\Phi(x,y,t) = \frac{72}{7} csc^2\left(\frac{i(x+y-4t)}{\sqrt{2}}\right).$$ |
| Taking $a = -\frac{1}{4}, b = \frac{1}{4}, r = 1, c = 4$ and $u(x,y,t) = \Phi(x,y,t)$, then the solution of Eq (29) turns to $$\Phi(x,y,t) = 12\, sech^2\left(\frac{x+y-4t}{\sqrt{2}}\right).$$ | Taking $\lambda = 1, \mu = 1, \gamma = 1, p = 1, r = 11, q = 1, \alpha = -\frac{1}{2}$ and $u_{46}(x,y,t) = \Phi(x,y,t)$, then the solution turns to $$\Phi(x,y,t) = 12\, sech^2\left(\frac{x+y-4t}{\sqrt{2}}\right).$$ |
| Taking $a = -\frac{1}{4}, b = \frac{1}{4}, r = 1, c = 4$ and $u(x,y,t) = \Phi(x,y,t)$, then the solution of Eq (28) turns to $$\Phi(x,y,t) = 12\, csch^2\left(\frac{x+y-4t}{\sqrt{2}}\right).$$ | Taking $\lambda = 1, \mu = 1, \gamma = 1, p = 1, r = 11, q = 1, \alpha = -\frac{1}{2}$ and $u_{47}(x,y,t) = \Phi(x,y,t)$, then the solution turns to $$\Phi(x,y,t) = 12\, csch^2\left(\frac{x+y-4t}{\sqrt{2}}\right).$$ |

method yields a substantially larger number of wave solutions compared to both the sine-cosine and tanh methods. In the current study, a unified scheme was applied to the specified model, revealing thirty-two solutions independently. Additionally, forty-six solutions were obtained from the KP-BBM model through the AAE scheme. It is important to note that the solutions derived in our research differ from those documented in Ref. [16].

**Remarks.** We have verified these solutions with Maple by putting them back into the original equation.

**Table 2. Comparison of our solutions and Wazwaz [16] solutions obtained by the tanh method.**

| Wazwaz [16] solutions obtained by the tanh method | Our solutions obtained by the AAE method |
|---|---|
| Taking $a = \frac{2}{3}, b = -2, r = 1, c = -2$ and $u(x,y,t) = \Phi(x,y,t)$, then the solution of Eq (38) turns to $$\Phi(x,y,t) = -3\left(1 + 3tan^2\left(\frac{x+y+2t}{2}\right)\right).$$ | Taking $\lambda = 1, \mu = 1, \gamma = 1, p = 1, r = -9, q = 3, \beta = 1, \alpha = \frac{1}{2}$ and $u_{56}(x,y,t) = \Phi(x,y,t)$, then the solution turns to $$\Phi(x,y,t) = -3\left(1 + 3tan^2\left(\frac{x+y+2t}{2}\right)\right).$$ |
| Choosing $a = \frac{2}{3}, b = -2, r = 1, c = -2$ and $u(x,y,t) = \Phi(x,y,t)$, then the solution of Eq (39) turns to $$\Phi(x,y,t) = -3\left(1 - 3cot^2\left(\frac{x+y+2t}{2}\right)\right).$$ | Choosing $\lambda = 1, \mu = 1, \gamma = 1, p = 1, r = -9, q = 3, \beta = 1, \alpha = \frac{1}{2}$ and $u_{57}(x,y,t) = \Phi(x,y,t)$, then the solution turns to $$\Phi(x,y,t) = -3\left(1 + 3cot^2\left(\frac{x+y+2t}{2}\right)\right).$$ |
| Taking $a = \frac{5}{12}, b = -1, r = 1, c = -2$ and $u(x,y,t) = \Phi(x,y,t)$, then the solution of the Eq (36) turns to $$\Phi(x,y,t) = -\frac{24}{5}\left(1 - 3tanh^2\left(\frac{x+y+2t}{\sqrt{2}}\right)\right).$$ | Taking $\lambda = 1, \mu = 1, \gamma = 1, p = 1, r = -9, q = 3, \beta = 1, \alpha = \frac{1}{2}$ and $u_{62}(x,y,t) = \Phi(x,y,t)$, then the solution turns to $$\Phi(x,y,t) = -\frac{24}{5}\left(1 - 3tanh^2\left(\frac{x+y+2t}{\sqrt{2}}\right)\right).$$ |
| Taking $a = \frac{5}{12}, b = -1, r = 1, c = -2$ and $u(x,y,t) = \Phi(x,y,t)$, then the solution of the Eq (37) turns to $$\Phi(x,y,t) = -\frac{24}{5}\left(1 - 3coth^2\left(\frac{x+y+2t}{\sqrt{2}}\right)\right).$$ | Taking $\lambda = 1, \mu = 1, \gamma = 1, p = 1, r = -9, q = 3, \beta = 1, \alpha = \frac{1}{2}$ and $u_{63}(x,y,t) = \Phi(x,y,t)$, then the solution turns to $$\Phi(x,y,t) = -\frac{24}{5}\left(1 - 3coth^2\left(\frac{x+y+2t}{\sqrt{2}}\right)\right).$$ |

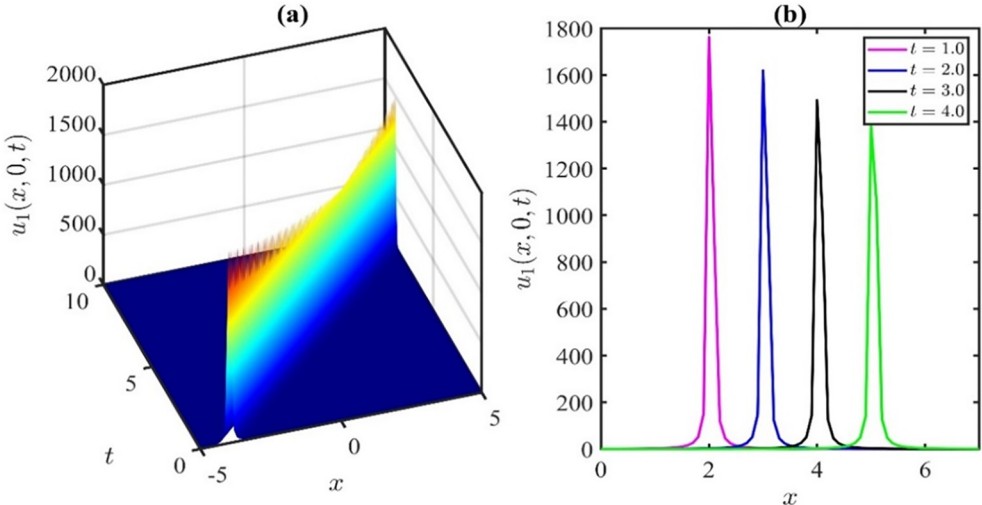

**Fig 1. Travelling wave profile of $u_1(x, y, t)$ for the values of $p = 1.0, q = 0.5, r = 0.01, k = -0.02, \mu = 0.01, \lambda = 0.21, H = 1, l = 0.01, \chi = 0.21$ and $y = 0$**

## 3. Graphical and physical explanations of the KP-BBM model

We will now offer insights into the dynamics of waves by investigating the impact of nonlinear parametric factors on the derived solutions. The significance of the nonlinear coefficient in the nonlinear KP-BBM equation lies in its crucial role in shaping the behaviour and evolution of solutions.

The 3D representation of the solution $u_1$ is depicted in Fig 1(A), while Fig 1(B) illustrates the associated behaviour of the travelling wave under specific parameter values: $p = 1.0, q = 0.5, r = 0.01, k = -0.02, \mu = 0.01, \lambda = 0.21, H = 1, l = 0.01, \chi = 0.21, y = 0$. In this representation, it is evident that the amplitude of the wave diminishes over time within the defined domain. Fig 2 elucidates the parametric influence of the nonlinear coefficient $p$. Fig 2(A) presents a 3D plot with varying $p$ at time $t = 2$, while Fig 2(B) provides the corresponding 2D representation. Analysis of the simulations depicted in Fig 2 leads us to infer that

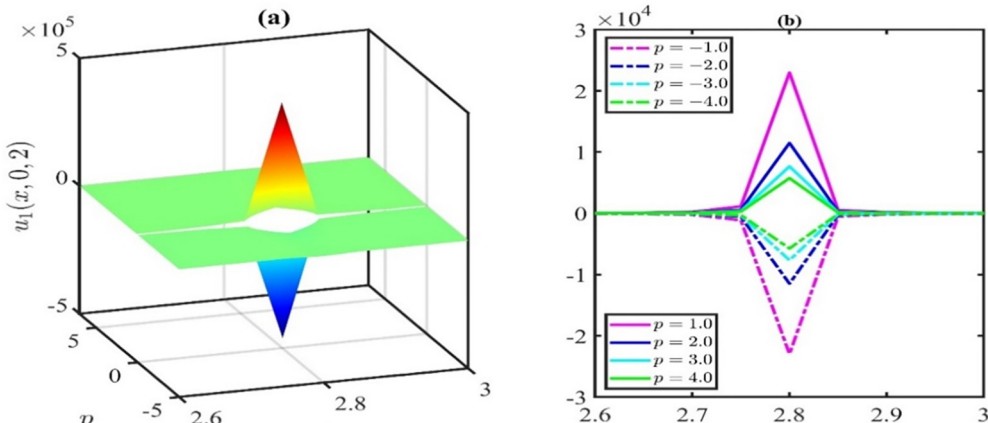

**Fig 2. Effect of nonlinear parameter $p$ on the solution $u_1(x, y, t)$ for the values of $t = 2, q = 0.3, r = 0.1, k = -0.02, \mu = 0.01, \lambda = 0.21, H = 0.1, l = 0.01, \chi = 0.21$ and $y = 0$.**

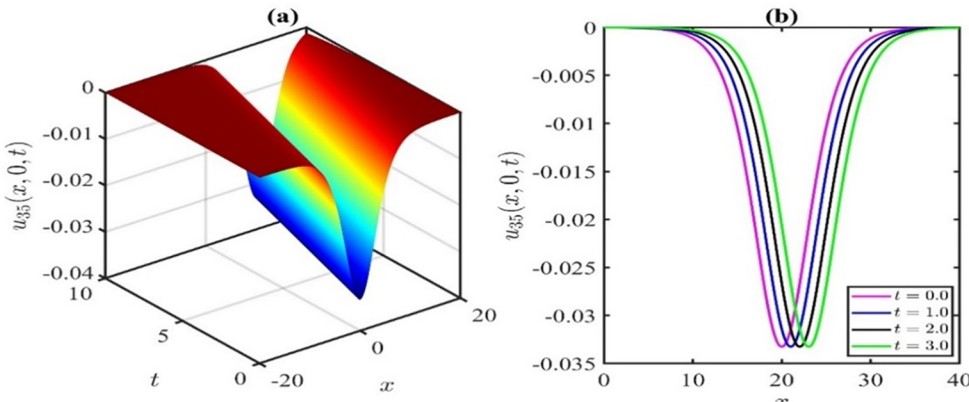

**Fig 3. Schematic illustration of dark soliton type amplitude of KP-BBM equation corresponds to the solution**
$u_{35}(x, y, t)$ **for the values of** $p = 1, \mu = 0.1, \alpha = 0.25, \beta = 1, \lambda = 0.5, \gamma = 0.1, q = 0.1, r = 0.2,$ **and** $y = 0.$

the wave amplitude increases as we deviate from $p = 0$ in either the positive or negative direction.

Figs 3 and 4 schematically illustrate two characteristic soliton profiles, denoted as the dark soliton (also recognized as gray and black solitons) and bright soliton, for the values of the $p = 1$ and $p = -1$, respectively, aligning with the solution $u_{35}(x,y,t)$. Simulations in Fig 3 were executed with the parameter values $p = 1, \mu = 0.1, \alpha = 0.25, \beta = 1, \lambda = 0.5, \gamma = 0.1,$ $q = 0.1, r = 0.2,$ and $y = 0$. In Fig 3(B), snapshots were acquired at $t = 0,1,2,3$, illustrating the propagation of the dark soliton in the positive direction of the $x$−axis. Subsequently, simulations in Fig 4 were performed for the parameter values of $p = -1, \mu = 0.1, \alpha = 0.25, \beta = 1, \lambda = 0.5, \gamma = 0.1, q = 0.1, r = 0.2,$ and $y = 0$. In Fig 4(B), snapshots were captured at $t = 0,1,2,3$, depicting the propagation of the bright soliton in the positive direction of the $x$ −axis.

The impact of the nonlinear coefficient p on the solution curve $u_{35}(x,y,t)$ is illustrated in Fig 5. The simulations presented in Fig 5(B) reveal that the influence of the nonlinear coefficient $p$ exhibits symmetry of wave amplitude about the horizontal axis. In this particular simulation, we systematically varied the parameter $p$ at time $t = 3$, while maintaining the values of

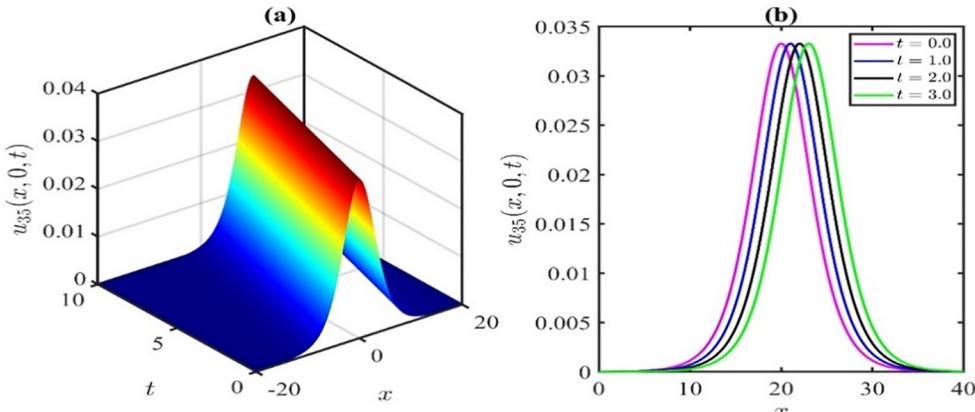

**Fig 4. Schematic illustration of bright soliton type amplitude of KP-BBM equation corresponds to the solution**
$u_{35}(x, y, t)$ **for the values of** $p = -1, \mu = 0.1, \alpha = 0.25, \beta = 1, \lambda = 0.5, \gamma = 0.1, q = 0.1, r = 0.2,$ **and** $y = 0.$

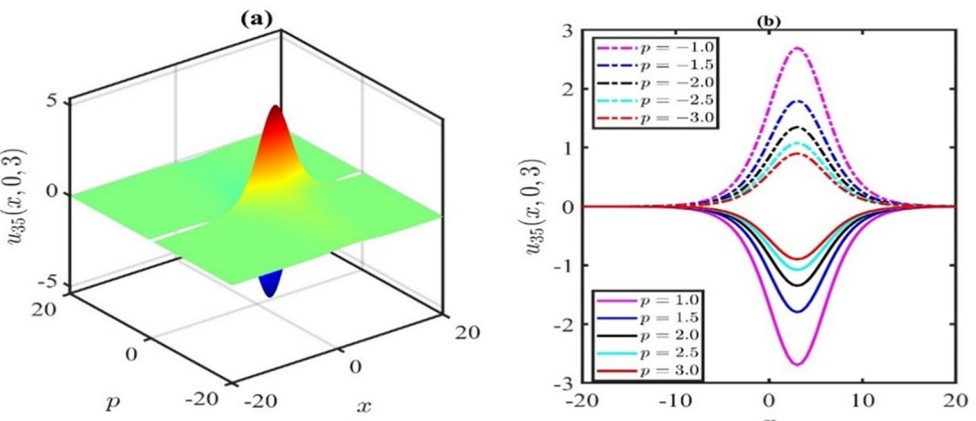

**Fig 5. Effects of nonlinear parameter $p$ on the solution $u_1(x, y, t)$ for the values of $t = 3, \mu = 0.1, \alpha = 0.25, \beta = 1, \lambda = 0.5, \gamma = 0.1, q = 0.1, r = 0.2$ and $y = 0$.**

$\mu = 0.1, \alpha = 0.25, \beta = 1, \lambda = 0.5, \gamma = 0.1, q = 0.1, r = 0.2$, and $y = 0$. Positive values of $p$ result in a dark soliton, while negative values of $p$ yield a bright soliton. However, when $p = 0$, singularities are stipulated. Based on the observations in Figs 2 and 5, it can be inferred that whenever a bright soliton occurs for $p = M$, a symmetrical dark soliton will emerge when $p = -M$ ($M$ being a positive number). For positive $p$, an escalation in the $p$ values leads to an augmentation in the wave amplitude. Conversely, for negative $p$, a reduction in the $p$ values increases the wave amplitude.

## 4. Stability analysis of the model

In this segment, we explore the stability analysis of the attained solutions through the application of a planar dynamical system. To facilitate this examination, we assume that the system described in Eq (2.2) can be represented in the following dynamical system as

$$
\begin{cases}
\dfrac{dX}{d\xi} = Y \\
\dfrac{dY}{d\xi} = \dfrac{p}{q\sigma\lambda}X^2 - \dfrac{\sigma\lambda - r\mu^2 - \lambda^2}{q\sigma\lambda^3}X
\end{cases}.
\tag{4.1}
$$

This system introduces the widely recognized phase portraits in $(X, Y)$-plane including parameters $p, q, \sigma, \lambda$ and $r$ that pertains to optical soliton solutions of the KP-BBM model. The differential equation specified in either Eq (2.2) or Eq (4.1) is derived from the corresponding Hamiltonian function by using the Hamilton canonical equations $X' = \frac{\partial H}{\partial Y}$ and $Y' = -\frac{\partial H}{\partial X}$ as

$$
H(X, Y) = \frac{Y^2}{2} + \frac{\sigma\lambda - r\mu^2 - \lambda^2}{2q\sigma\lambda^3}X^2 - \frac{p}{3q\sigma\lambda}X^3.
\tag{4.2}
$$

Now, the three equilibrium points of (4.1) are (0,0) and $\left(\frac{\sigma\lambda - r\mu^2 - \lambda^2}{p\lambda^2}, 0\right)$, implies that $p, \lambda \neq 0$. The characteristics equation of the Jacobian matrix is given by

$$
\psi^2 - \frac{2p}{q\sigma\lambda}X + \frac{\sigma\lambda - r\mu^2 - \lambda^2}{q\sigma\lambda^3} = 0.
$$

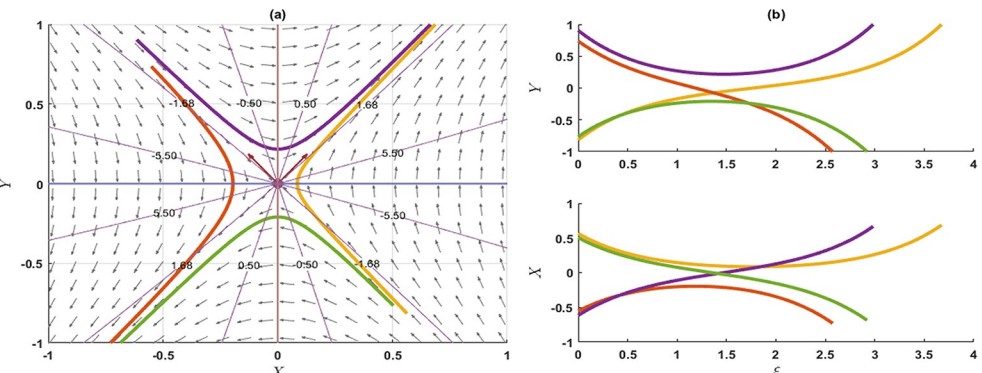

**Fig 6. The phase portrait and associated solution of the planar dynamical system (4.1) are presented for selected parameters as $p = 1, q = 0.3, \sigma = 0.02, \lambda = 2, r = 0.1, \mu = 0.03$.** The equilibrium point $(0, 0)$ is an unstable saddle.

**Stability of the equilibrium point (0,0):** This time, the characteristics roots are $\psi_1 = \pm i\sqrt{\frac{\sigma\lambda - r\mu^2 - \lambda^2}{q\sigma\lambda^3}}$ and $\psi_2 = \pm i\sqrt{\frac{\sigma\lambda - r\mu^2 - \lambda^2}{q\sigma\lambda^3}}$, such that $p,\sigma,\lambda \neq 0$. If $\frac{\sigma\lambda - r\mu^2 - \lambda^2}{q\sigma\lambda^3} > 0$, then the eigenvalues $\psi_1$ and $\psi_2$ are the imaginary. So, the equilibrium point (0,0) is a stable centre or ellipse. If $\frac{\sigma\lambda - r\mu^2 - \lambda^2}{q\sigma\lambda^3} < 0$, then the eigenvalues $\psi_1$ and $\psi_2$ are real and opposite signs and the given equilibrium point is an unstable saddle point. As a result of this analysis, it is evident that the equilibrium point can be characterized as an unstable saddle point, as indicated in Figs 6, 7 and 9. Conversely, the specified point exhibits an elliptical shape and represents a stable centre, as illustrated in Fig 8.

**Stability of the equilibrium point $\left(\frac{\sigma\lambda - r\mu^2 - \lambda^2}{p\lambda^2}, 0\right)$:** This time, the characteristic roots are $\psi_1 = \frac{\sigma\lambda - r\mu^2 - \lambda^2}{q\sigma\lambda^3}$ and $\psi_2 = -\frac{\sigma\lambda - r\mu^2 - \lambda^2}{q\sigma\lambda^3}$, such that $q,\sigma,\lambda \neq 0$. If $\frac{\sigma\lambda - r\mu^2 - \lambda^2}{q\sigma\lambda^3} > 0$, then the eigenvalues $\psi_1$ and $\psi_2$ are real and opposite signs. So, the equilibrium points $\left(\frac{\sigma\lambda - r\mu^2 - \lambda^2}{p\lambda^2}, 0\right)$ are unstable saddle points. On the other hand, $\frac{\sigma\lambda - r\mu^2 - \lambda^2}{q\sigma\lambda^3} < 0$, then the eigenvalues $\psi_1$ and $\psi_2$ are imaginary. Therefore, the given equilibrium points are the stable centre. As a result of this analysis, it is evident

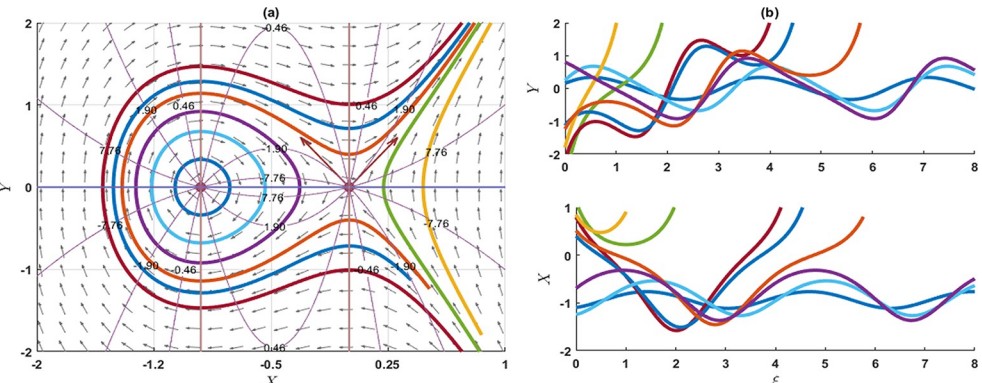

**Fig 7. The phase portrait and associated solution of the planar dynamical system (4.1) are presented for selected parameters as $p = 1, q = 0.05, \sigma = 0.2, \lambda = 0.1, r = 0.01, \mu = 0.3$.** The equilibrium point (0,0) is an unstable saddle, while the equilibrium point at (−0.95,0) is characterized as a centre.

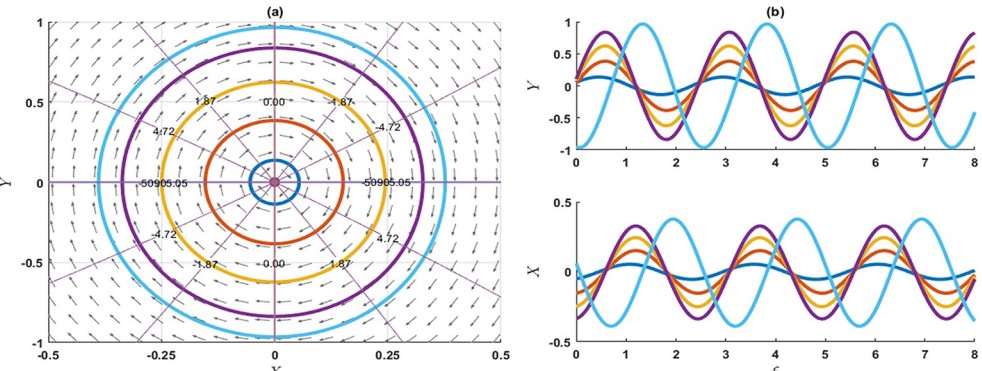

**Fig 8. The phase portrait and associated solution of the planar dynamical system (4.1) are presented for selected parameters as $p = 2, q = 0.1, \sigma = -0.01, \lambda = 2, r = 0.01, \mu = 0.03$.** The equilibrium point (0,0) is characterized as a centre.

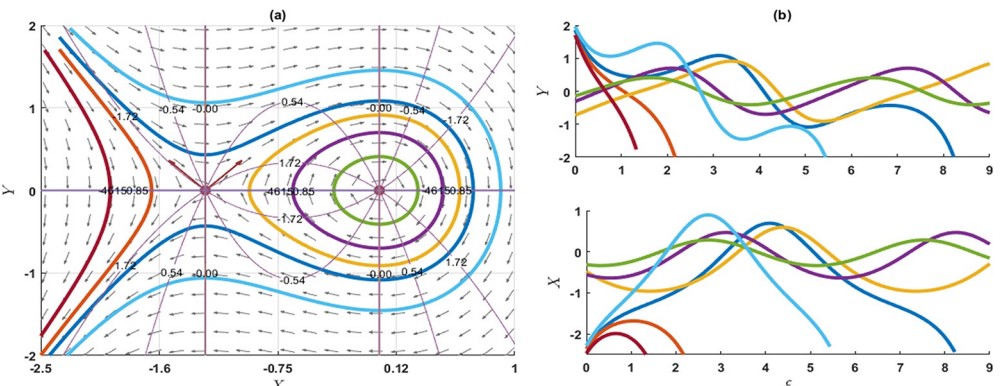

**Fig 9. The phase portrait and associated solution of the planar dynamical system (4.1) are presented for the selected parameter as $p = 0.7, q = 0.1, \sigma = 0.2, \lambda = -1, r = 0.01, \mu = -0.3$.** The equilibrium point (0,0) is identified as a centre, while the equilibrium point at (−1.28,0) is characterized as an unstable saddle.

that the equilibrium point can be characterized as an unstable saddle point, as indicated in Figs 6, 7 and 9. Conversely, the specified point exhibits an elliptical shape and represents a stable centre, as illustrated in Fig 8.

## 5. Conclusion

In this investigation, we have successfully obtained precise wave solutions for the KP-BBM model using a direct approach through both the unified and AAE methods. A comparative analysis with existing literature has uncovered a diverse array of solutions, each characterized by distinct behaviours. These newly derived solutions are unprecedented and hold significant promise for addressing real-world challenges associated with the KP-BBM model in diverse physics and engineering domains. Notably, these innovative exact wave solutions have the potential to make substantial contributions to fields such as fluid dynamics, ocean engineering, and applied mathematics. The employed techniques demonstrate robustness and high efficiency. Furthermore, we conducted a bifurcation analysis of the model, assessing the stability of equilibrium points. The resulting phase portrait of the model is depicted in Figs 6–9.

Additionally, 2D combined and 3D plots are presented for visually representing the solutions, facilitating the comprehension of wave motions. This research delves into the intricate dynamics of the KP-BBM equation, particularly focusing on the parameter $p$ and its influence on soliton formations. Our findings indicate that variations in parameter values can induce shifts in the dynamics of soliton solutions within the KP-BBM model. The comparative analysis of the solutions of the KP-BBM model through the AAE method and unified method reveals significant findings. Our assertion is that the AAE method and unified method surpass the sinecosine method and the tanh method, as demonstrated by the significantly greater number of wave solutions they produce. It is imperative to emphasize that our research findings diverge from those documented in Ref. [16].

Our comprehensive exploration of soliton dynamics and the obtained solutions not only enhances the understanding of the KP-BBM equation but also underscores the efficacy of the AAE method and unified method in producing a myriad of wave solutions, holding substantial potential for applications in various physics and engineering domains. In summary, both the unified scheme and the AAE scheme prove to be potent, compatible, and straightforward methods for deriving comprehensive wave solutions with various free parameters, offering valuable insights into wave profiles across different scenarios.

## Supporting information

**S1 File. Unified method.**
(DOCX)

**S2 File. Advanced auxilairy equation method.**
(DOCX)

## Acknowledgments

The authors would like to thank the editor of the journal and anonymous reviewers for their generous time in providing detailed comments and suggestions that helped us to improve the paper.

## Author Contributions

**Conceptualization:** S. M. Rayhanul Islam.

**Data curation:** S. M. Rayhanul Islam.

**Formal analysis:** Kamruzzaman Khan.

**Investigation:** S. M. Rayhanul Islam, Kamruzzaman Khan.

**Methodology:** S. M. Rayhanul Islam.

**Software:** S. M. Rayhanul Islam, Kamruzzaman Khan.

**Supervision:** Kamruzzaman Khan.

**Validation:** Kamruzzaman Khan.

**Visualization:** Kamruzzaman Khan.

**Writing – original draft:** S. M. Rayhanul Islam, Kamruzzaman Khan.

**Writing – review & editing:** S. M. Rayhanul Islam, Kamruzzaman Khan.

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
