## [Decision Letter · Decision Letter 0]

24 Jan 2024

PONE-D-23-40797Investigating wave solutions and impact of nonlinearity: Comprehensive study of the KP-BBM model with bifurcation analysisPLOS ONE

Dear Dr. Islam S. M. Rayhanul,

Thank you for submitting your manuscript to PLOS ONE. After careful consideration, we feel that it has merit but does not fully meet PLOS ONE’s publication criteria as it currently stands. Therefore, we invite you to submit a revised version of the manuscript that addresses the points raised during the review process.

We look forward to receiving your revised manuscript.

Kind regards,

Muhammad Aqeel, Ph.D

Academic Editor

PLOS ONE

Journal Requirements:

 Whilst you may use any professional scientific editing service of your choice, PLOS has partnered with both American Journal Experts (AJE) and Editage to provide discounted services to PLOS authors. Both organizations have experience helping authors meet PLOS guidelines and can provide language editing, translation, manuscript formatting, and figure formatting to ensure your manuscript meets our submission guidelines. To take advantage of our partnership with AJE, visit the AJE website (http://aje.com/go/plos) for a 15% discount off AJE services. To take advantage of our partnership with Editage, visit the Editage website (www.editage.com) and enter referral code PLOSEDIT for a 15% discount off Editage services. If the PLOS editorial team finds any language issues in text that either AJE or Editage has edited, the service provider will re-edit the text for free.

 A clean copy of the edited manuscript (uploaded as the new *manuscript* file)”"

4. In the online submission form, you indicated that [Insert text from online submission form here].

Additional Editor Comments (if provided):

One reviewer has major points that should be addressed in detail please.

Reviewers' comments:

Reviewer's Responses to Questions

**Comments to the Author**

1. Is the manuscript technically sound, and do the data support the conclusions?

Reviewer #1: Yes

Reviewer #2: Yes

Reviewer #3: Yes

2. Has the statistical analysis been performed appropriately and rigorously? 

Reviewer #1: Yes

Reviewer #2: Yes

Reviewer #3: Yes

3. Have the authors made all data underlying the findings in their manuscript fully available?

Reviewer #1: Yes

Reviewer #2: Yes

Reviewer #3: Yes

4. Is the manuscript presented in an intelligible fashion and written in standard English?

Reviewer #1: Yes

Reviewer #2: Yes

Reviewer #3: Yes

5. Review Comments to the Author

Reviewer #1: In this article, authors have explored the (2+1)-dimensional Kadomtsev-Petviashvili-Benjamin-Bona Mahony equation using two effective methods via unified and advanced auxiliary equation schemes to derive precise wave solutions. The obtained numerical results are reliable, straightforward, and potent tools for analyzing various nonlinear evolution equations. The article is interesting and is on average level. However, I suggest some necessary corrections in attachment.

Reviewer #2: Investigating Wave Solutions and Impact of Nonlinearity: comprehensive Study of the KP-BBM Model with Bifurcation Analysis

Summary and the overall:

Well written introduction, mathematical background, literature review, objectives, mathematical analysis, results and conclusion. The presentation of the concepts of the article was smooth and sequential.

Major comments:

1-Aim: The aim of the study is clearly states. The study focused on applying the unified and Advanced Auxiliary Equation (AAE) methods to find soliton solutions of the KP-BBM model . The study provided an analysis of the impact of some parameters and ended with a stability analysis on the model.

2- Method: The innovation in this study was to provide solutions to the KP-BBM model using methods that have not been utilized in literature, namely the unified and the AAE methods using Maple software.

4- Discussion and Applications:

The discussion was well written and organized. The study presents good computation work. The paper revealed 32 solutions to the KP-BBM model using the unified method and 46 solutions using the AAE method. The problem, however, is that none of the solutions presented by the paper was obtained earlier in the literature. This causes doubt about the accuracy of the solutions. I would suggest that the authors should investigate solutions under the same conditions on the parameters as was discussed in the literature to enable the comparison between the obtained solutions and existing solutions as stated in section 1.5. I also suggest comparing the stability of the solutions obtained and existing solutions in literature.

Reviewer #3: The manuscript is technically sound full. It is presented written in an intelligent fashion and written in a standard English. The way the writers try to show the gap is outstanding. The conclusions are drawn appropriately based on the data presented.

Besides this, I have some comments 1) Their is no Acronym section. It is mandatory to insert this section unless no one knows what does it mean KPBBM, AAE, etc. 2) From your objectives I didn't see clearly what is your general objective and your specific objectives. So, try to show clearly 3) From your back ground on page 2 line 62 you try to cite [16-29] but from your reference section their is no reference number 28 and 29. So, what does it mean?

Generally, If the above comments are corrected I suggest to accept this manuscript.

6. PLOS authors have the option to publish the peer review history of their article (what does this mean?). If published, this will include your full peer review and any attached files.

Reviewer #1: **Yes: **Dr. Muhammad Haroon Aftab

Reviewer #2: No

Reviewer #3: No

---

## [Author Response · Author response to Decision Letter 0]

3 Feb 2024

Response to Reviewers Comments

Manuscript Number: PONE-D-23-40797

Manuscript title: Investigating wave solutions and impact of nonlinearity: Comprehensive study of the KP-BBM model with bifurcation analysis

The authors appreciate and thank the Editor/Reviewer’s for his/her precious time, careful reading and making useful comments/suggestions towards the manuscript. The manuscript has been revised keeping in view of all the comments and suggestions made by the Editor/Reviewer’s. In the revised manuscript, changes have been highlighted with red colour. The author’s responses to the Editor/Reviewer’s comments are listed below:

Reviewer 1:

In this article, authors have explored the (2+1)-dimensional Kadomtsev-Petviashvili-Benjamin-Bona Mahony equation using two effective methods via unified and advanced

auxiliary equation schemes to derive precise wave solutions. The obtained numerical results are reliable, straightforward, and potent tools for analyzing various nonlinear evolution equations. The article is interesting and is on average level. However, I suggest some necessary corrections as follows:

Reviewer Comment #1: “What are the applications of this work?”

Authors’ Reply: Thank you very much for your nice comments. The applications of this work are mentioned in below the sub-section 1.1 (Background and mathematical model). Please see in the revise manuscript.

Reviewer Comment #2: “Add statement of novelty.”

Authors’ Reply: Thank you very much for your nice comments. Please see in sub-section 1.4 in the revise manuscript.

Reviewer Comment #3: “3- P-1 (+14): replace “figures” with “Figures”.”

Authors’ Reply: Please see the revise manuscript.

Reviewer Comment #4: “Figures (6-9) are not cited.”

Authors’ Reply: Thank you very much for your kind suggestion. We studied the stability analysis of the solutions from the stated model and Figures (6-9) are phase portrait of the model, which are mention in stability analysis of the model and conclusion sections. 

Reviewer Comment #5: “Improve the resolution of all Figures.”

Authors’ Reply: Thank you for your valuable suggestion. We improved the resolution of all Figures and please see in the revise manuscript.

Reviewer Comment #6: “On line 382, there should be space after “equilibrium point.”

Authors’ Reply: Thank you very much for your kind suggestion. Yes, corrected.

Reviewer 2:

Summary and the overall: Well written introduction, mathematical background, literature review, objectives, mathematical analysis, results and conclusion. The presentation of the concepts of the article was smooth and sequential.

Reviewer Comment #1: “1-Aim: The aim of the study is clearly states. The study focused on applying the unified and Advanced Auxiliary Equation (AAE) methods to find soliton solutions of the KP-BBM model. The study provided an analysis of the impact of some parameters and ended with a stability analysis on the model.”

Authors’ Reply: Thank you very much for your kind comment. Please see in the revise manuscript.

Reviewer Comment #2: “2- Method: The innovation in this study was to provide solutions to the KP-BBM model using methods that have not been utilized in literature, namely the unified and the AAE methods using Maple software.”

Authors’ Reply: Thank you very much for your kind comments. Please see in Appendix A and B in the revise manuscript.

Reviewer Comment #3: “4- Discussion and Applications: The discussion was well written and organized. The study presents good computation work. The paper revealed 32 solutions to the KP-BBM model using the unified method and 46 solutions using the AAE method. The problem, however, is that none of the solutions presented by the paper was obtained earlier in the literature. This causes doubt about the accuracy of the solutions. I would suggest that the authors should investigate solutions under the same conditions on the parameters as was discussed in the literature to enable the comparison between the obtained solutions and existing solutions as stated in section 1.5. I also suggest comparing the stability of the solutions obtained and existing solutions in literature.”

Authors’ Reply: Thank you very much for your nice comment. I have compared between our solutions and Wazwaz [16] solutions in sub-section 2.3. Please in the revise manuscript.

Reviewer 3:

The manuscript is technically sound full. It is presented written in an intelligent fashion and written in a standard English. The way the writers try to show the gap is outstanding. The conclusions are drawn appropriately based on the data presented. Besides this, I have some comments 

Reviewer Comment #1: “1) There is no Acronym section. It is mandatory to insert this section unless no one knows what does it mean KPBBM, AAE, etc.”

Authors’ Reply: Yes, Added in below the abstract and keywords in the revise manuscript. 

Reviewer Comment #2: “2) From your objectives I didn't see clearly what is your general objective and your specific objectives. So, try to show clearly”

Authors’ Reply: Thank you very much for your kind comments. Please see in the revise manuscript in sub-section 1.4 (Aim and Objectives).

Reviewer Comment #3: “3) From your back ground on page 2 line 62 you try to cite [16-29] but from your reference section their is no reference number 28 and 29. So, what does it mean?”

Authors’ Reply: This is typos error in original manuscript, but two new references added in the revise manuscript.

Finally, we greatly appreciate your suggestion and valuable comments on our paper. We hope our revisions are good enough to be published in this journal.

Sincerely yours,

S M Rayhanul Islam

Corresponding author

---

## [Decision Letter · Decision Letter 1]

28 Feb 2024

Investigating wave solutions and impact of nonlinearity: Comprehensive study of the KP-BBM model with bifurcation analysis

PONE-D-23-40797R1

Dear Dr. Islam S. M. Rayhanul

We’re pleased to inform you that your manuscript has been judged scientifically suitable for publication and will be formally accepted for publication once it meets all outstanding technical requirements.

Kind regards,

Muhammad Aqeel, Ph.D

Academic Editor

PLOS ONE

Additional Editor Comments (optional):

Accept

Reviewers' comments:

Reviewer's Responses to Questions

**Comments to the Author**

1. If the authors have adequately addressed your comments raised in a previous round of review and you feel that this manuscript is now acceptable for publication, you may indicate that here to bypass the “Comments to the Author” section, enter your conflict of interest statement in the “Confidential to Editor” section, and submit your "Accept" recommendation.

Reviewer #1: All comments have been addressed

Reviewer #2: All comments have been addressed

Reviewer #3: All comments have been addressed

2. Is the manuscript technically sound, and do the data support the conclusions?

Reviewer #1: Yes

Reviewer #2: Yes

Reviewer #3: Yes

3. Has the statistical analysis been performed appropriately and rigorously? 

Reviewer #1: Yes

Reviewer #2: Yes

Reviewer #3: Yes

4. Have the authors made all data underlying the findings in their manuscript fully available?

Reviewer #1: Yes

Reviewer #2: Yes

Reviewer #3: Yes

5. Is the manuscript presented in an intelligible fashion and written in standard English?

Reviewer #1: Yes

Reviewer #2: Yes

Reviewer #3: Yes

6. Review Comments to the Author

Reviewer #1: Tile: Investigating wave solutions and impact of nonlinearity: Comprehensive study of the KP-BBM model with bifurcation analysis:

All suggestions have been implemented accordingly.

Reviewer #2: The study presents good computation work. The paper revealed 32 solutions to the KP-BBM model using the unified method and 46 solutions using the AAE method. The authors provided a comparison study of their solutions and the solutions presented in the literature which supports the accuracy of the used method.

Reviewer #3: (No Response)

7. PLOS authors have the option to publish the peer review history of their article (what does this mean?). If published, this will include your full peer review and any attached files.

Reviewer #1: **Yes: **Dr. Muhammad Haroon Aftab

Reviewer #2: No

Reviewer #3: No

---

## [Editor Report · Acceptance letter]

4 Apr 2024

PONE-D-23-40797R1 

PLOS ONE

Dear Dr. Rayhanul Islam, 

I'm pleased to inform you that your manuscript has been deemed suitable for publication in PLOS ONE. Congratulations! Your manuscript is now being handed over to our production team.

Kind regards, 

on behalf of

Dr. Muhammad Aqeel 

Academic Editor

PLOS ONE